# Erasing and Tampering Statistical Watermarks via Re-watermarking in Large Language Models

## Abstract

The rapid development and widespread adoption of large language models have intensified concerns about copyright disputes, misinformation spread, and content authenticity. Statistical watermarking has been proposed as a potential solution for content source verification, though its reliability remains questionable. This study examines a re-watermarking attack based on text rephrasing. Our theoretical analyses and experimental results demonstrate that: (1) new watermarks can be successfully applied to already watermarked text; (2) these new watermarks effectively overwrite the originals, making them undetectable; and (3) compared to existing rephrasing-only attacks, re-watermarking causes comparable degradation in text fidelity. These findings reveal significant vulnerabilities in statistical watermarking techniques, challenging their effectiveness as reliable mechanisms for content attribution.

## 1 Introduction

The rapid expansion of large language models has significantly advanced the capabilities of AI-generated text, allowing these models to produce content that closely mimics human writing. However, this rapid development also raises substantial concerns. Distinguishing between AI-generated and human-generated text, as well as identifying the sources of AI-generated content, presents challenges including copyright infringement, privacy violations, and the spread of misinformation. These concerns have prompted responses from regulatory bodies at both national and international levels.

**Related Works:** To trace the origin of content generated by large language models, watermarking techniques have become a widely studied solution. A significant category of watermarking methods, termed Statistical Watermarking (Huang et al. (2023)), introduces statistical signals into the text generation process, enabling watermark verification to be formalized as a hypothesis testing problem. Among such methods, the Green/Red List watermark (Kirchenbauer et al. (2023)) partitions the vocabulary into green and red lists, encouraging token selection from the green list during decoding. This approach has inspired various refinements to enhance robustness and applicability (Zhao et al. (2023); Kirchenbauer et al.; Wu et al.). Another example is the Gumbel Watermark, based on exponential minimal sampling, first introduced by Aaronson & Kirchner (2023) and extended in Zhao et al. (2024) and Fu et al. (2024). Additionally, some methods leverage inverse transform sampling to add the watermark, where the next-token generation first samples a pseudo-random number from a uniform distribution between 0 and 1, then maps this number to a token. The mapping process is designed to ensure that when the pseudo-random number is truly random, the sampling remains unbiased (Kuditipudi et al. (2024); Christ et al. (2024); Hu et al.; Li et al. (2024)). Notably, recent studies have also explored an unbiased version of the Green/Red List watermark, aiming to reduce its impact on the quality of generated text while maintaining detectability (Xie et al. (2024)).

Although statistical watermarking has been widely studied and various versions exist, its reliability remains a subject of debate. On one hand, some studies support its robustness. For example, Kirchenbauer et al. investigated an improved watermarking method based on the Green/Red List based approach and evaluated its resilience under different attack scenarios, including machine rephrasing (using trained paraphrasing models such as Dipper (Krishna et al. (2024)) or GPT) and human rephrasing. Experimental results indicate that as the length of the paraphrased text increases, the detection rate of the watermark after the attack

significantly improves. This suggests that for longer texts, the watermark can still maintain a certain level of robustness. On the other hand, some studies have challenged the robustness of statistical watermarking by designing various attack strategies, leading to opposite conclusions. For example, Zhang et al. indicates that if the attacker has access to the watermark detector, they can always remove the watermark while preserving the quality of the text by performing a sufficient number of repeat rephrasings and selecting the highest-quality responses. Pang et al. (2024) explores the use of repeated queries to recover the original token distribution, effectively removing the watermark, especially in scenarios where multiple keys are used. Besides, some studies have explored how to reverse-engineer a watermark and then remove or spoof it. For instance, Jovanović et al. presents a method for inferring Green List tokens in the Green List/Red List watermarking scheme by scoring candidate tokens. The key idea is to leverage a reference model to determine whether a token appears 'normal' given a specific prefix. Sadasivan et al. (2023) also explores learning Green List tokens by repeatedly querying the model to estimate the token distribution. Gu et al. explores the approach of training a student model to replicate the behavior of a watermarked model.

However, the effectiveness of existing attack methods in compromising watermarks remains limited. Rephrase attacks, for instance, often require multiple iterations to effectively remove the watermark (Zhang et al.), and their success rate decreases when applied to longer texts (Kirchenbauer et al.). Similarly, reverse-engineering-based attacks typically require a large number of queries to infer the watermarking scheme. For example, Pang et al. (2024) notes that Sadasivan et al. (2023) issued one million queries to analyze token distributions and extract the watermarking scheme. Likewise, Jovanović et al. leveraged tens of thousands of responses, each containing a few hundred tokens, to extract the watermark, while Gu et al. utilized 640,000 watermarked samples, each with a length of 256 tokens, for watermark distillation.

Recently, Luo et al. (2024) proposed a rewatermarking attack, which injects a new watermark into previously watermarked text to suppress the old watermark. However, they reported limited success. Their findings highlight two main limitations: (1) rewatermarking tends to cause significant degradation in text similarity, raising concerns about fidelity; and (2) in many cases, the presence of the original watermark appears to interfere with the application of the new one, which seems to instead suggest the robustness of the original watermarking scheme.

In this work, we revisit the rewatermarking attack paradigm and present a different perspective. By pairing watermarking with a high-quality rephraser, we demonstrate that rewatermarking can be a highly effective and practical attack. Specifically, we show that:

- Adversarial control over watermarked text: Our method allows an adversary to inject a new watermark during rephrasing, thereby altering the provenance signal and undermining watermark-based attribution.

- Increased attack success rate: Compared to conventional rephrasing—which only partially removes the original watermark—rewatermarking provides a stronger obfuscation effect, making the original watermark significantly harder to detect.

- Minimal impact on text fidelity: Empirical results show that our approach introduces negligible semantic degradation. To quantify this, we conducted a human evaluation study with undergraduate annotators who compared rewatermarked text with its original counterpart. Results indicate that in 84% of cases, the text pairs were judged to be semantically similar.

These findings offer a contrasting observation to that of Luo et al. (2024), and highlight a notable limitation in current statistical watermarking schemes: the presence of an existing watermark does not prevent the injection of a new one. In particular, we find that when combined with a strong rephraser, the new watermark can significantly reduce the detection rate of the original watermark while incurring only minimal loss in semantic similarity. This enables adversaries to rewrite the attribution signal while maintaining high text quality, thereby making provenance tracking more ambiguous and less reliable. Moreover, rewatermarking is both low-cost and query-efficient, challenging prevailing assumptions about the robustness of watermarking techniques.

## 1.1 Preliminaries

Before presenting our proposed attacks, we first provide some background on language models and watermarking.

Let $\mathcal{W}$ be a discrete vocabulary set representing a finite collection of tokens and . A *language model* is a function $p : \mathcal{W}^* \to \Delta(\mathcal{W})$ that maps a sequence of tokens of arbitrary length to a probability distribution over $\mathcal{W}$. Specifically, given any prefix $x \in \mathcal{W}^*$, the model defines a conditional probability distribution $p(\cdot \mid x)$ over the next token, indicating the likelihood of each possible continuation.

A watermarking mechanism $S$ is a decoding function that takes a probability distribution over all tokens, $P \in \Delta W$, along with a secret key $\xi$ as input, and produces a modified probability distribution $\widehat{P} = S(P, \xi)$ as output.

Given a language model $p$ and a watermarking mechanism $W$, we follow Li et al. (2024) to describe the text generation process.

The generation process proceeds as follows: Given a prefix $w_{1:t-1}$, which includes both the prompt and all previously generated tokens, the language model produces the next-token prediction (NTP) distribution:

$$P_t = p(\cdot \mid w_{1:t-1}),$$

where $P_t$ represents the probability of each token in the vocabulary being selected as the next token.

Once the NTP distribution $P_t$ is established, the decoder $\mathcal{S}$ samples the next token $w_t$ from the distribution $\mathcal{S}(P_t, \xi_t)$:

$$w_t \sim \widehat{P}_t = \mathcal{S}(P_t, \xi_t).$$

This token is then appended to the sequence, and the process repeats until a stop token is generated or a predefined sequence length is reached.

In the following, we introduce two widely used watermarking methods, which serve as the main focus of our attack analysis.

The first is the green/red list watermark, initially proposed in Kirchenbauer et al. (2023) and later refined in Zhao et al. (2023); Kirchenbauer et al..

**Definition 1** (Green/Red List Watermark)**.** *In the Green/Red List watermarking scheme, we define a deterministic mapping $h(w, \xi)$ that associates a token $w$ and a secret key $\xi$ to the set $\{0, 1\}$. This function serves as an indicator of whether a token belongs to a predefined list:*

- *$h(w, \xi) = 1$ if $w$ belongs to the **Green List**, meaning its probability will be enhanced.*

- *$h(w, \xi) = 0$ if $w$ belongs to the **Red List**, meaning its probability will not be enhanced.*

*Given an original token distribution $P \in \Delta(\mathcal{W})$, the modified distribution $\widehat{P} = S(P, \xi)$ incorporating the watermark is defined as:*

$$\widehat{P}(w) = \frac{P(w)e^{h(w, \xi)\delta}}{\sum_{w \in \mathcal{V}} P(w)e^{h(w, \xi)\delta}},$$

*where $\delta$ is a hyperparameter that controls the strength of the watermark.*

The second approach is the **Gumbel watermark**, initially proposed in Aaronson & Kirchner (2023) and later refined in Zhao et al. (2024); Fu et al. (2024).

**Definition 2** (Gumbel Watermark)**.** *In the Gumbel watermark, we define a secret key as a vector*

$$\xi = (\xi[i])_{i \in \mathcal{W}} \in [0, 1]^{|\mathcal{W}|}.$$

*Additionally, we define an ordering function*

$$\pi : \mathcal{W} \to \{1, 2, \ldots, |\mathcal{W}|\},$$

*which is a bijective mapping from the token set $\mathcal{W}$ to the index set $\{1, 2, \ldots, |\mathcal{W}|\}$.*

*Given the secret key $\xi$ and a probability distribution $P \in \Delta(\mathcal{W})$ over the token set, the modified distribution $\widehat{P} = S(P, \xi)$ is defined by*

$$\widehat{P}(w) = \begin{cases} 1, & \text{if } w = \arg \max_{v \in \mathcal{W}} \left( \xi[\pi(v)] \right)^{\frac{1}{P(v)}} \\ 0, & \text{otherwise.} \end{cases} \tag{1}$$

To detect the presence of a watermark, a common approach is to define a *score function $T$* that takes as input a token sequence $\{w_t\}$, the corresponding secret key $\{\xi_t\}$, and optionally the original next-token prediction distributions $\{P_t\}$. The function $T$ outputs a real-valued score $c$, which is then used to evaluate the confidence that the given text contains the watermark. Usually, $T$ is defined such that higher scores indicate a greater likelihood of the text being watermarked.

Specifically, for the Green/Red List Watermark, the score function $T$ is computed as

$$T(\{w_t\}, \{\xi_t\}) = 2 \cdot \frac{\left( \sum_{t=1}^{n} h(w_t, \xi_t) - \frac{n}{2} \right)}{\sqrt{n}},$$

where $n$ is the length of the sequence $\{w_t\}$.

Meanwhile, for the Gumbel Watermark, the score function $T$ is given by

$$T(\{w_t\}, \{\xi_t\}) = \sum_{t=1}^{n} \ln \left( \frac{1}{1 - \xi_t[\pi(w_t)]} \right).$$

Then, a typical detection mechanism $D : \mathcal{W}^\star \times \Xi^\star \to \{0, 1\}$ applies a threshold $\tau$ to the score $T$. Specifically, we define:

$$D(\{w_t\}, \{\xi_t\}) = \mathbf{1}\Big[ T(\{w_t\}, \{\xi_t\}) \geq \tau \Big],$$

where $\mathbf{1}[\cdot]$ is the indicator function. If $T$ exceeds $\tau$, the text is detected as watermarked.

## 2 A Simple Re-Watermarking Method to Overwrite Statistical Watermarks

We first recap a conceptually simple method to disrupt statistical watermarks, originally proposed by Luo et al. (2024): Rather than relying on learning-based or repeated generation-based strategies, the approach overwrites the original watermark by re-watermarking the text with a randomly selected key. Specifically, given a watermarked text, we input it into a large language model and prompt it to perform a synonym-based rephrase. During this process, we embed a new watermark, which can be generated using either the same family of watermarking methods or an alternative approach. Importantly, we always employ a newly generated set of pseudo-random numbers $\{\widehat{\xi_t}\}$ instead of the original ones. The final output is a token sequence with the newly embedded watermark, denoted as $\{\widehat{w}_t\}$. The details of this process are summarized in Algorithm 1.

In the following, we provide a theoretical explanation for why rewatermarking can effectively suppress the original watermark.

### 2.1 Theoretical Justification and Intuition

We present a theorem to illustrate why the rewatermark method can be effective under idealized conditions. Although the simplified theoretical assumptions used in these theorems may not entirely reflect real-world complexities, they aim to provide intuition on how the proposed rewatermark attack can remove or obscure existing watermarks.

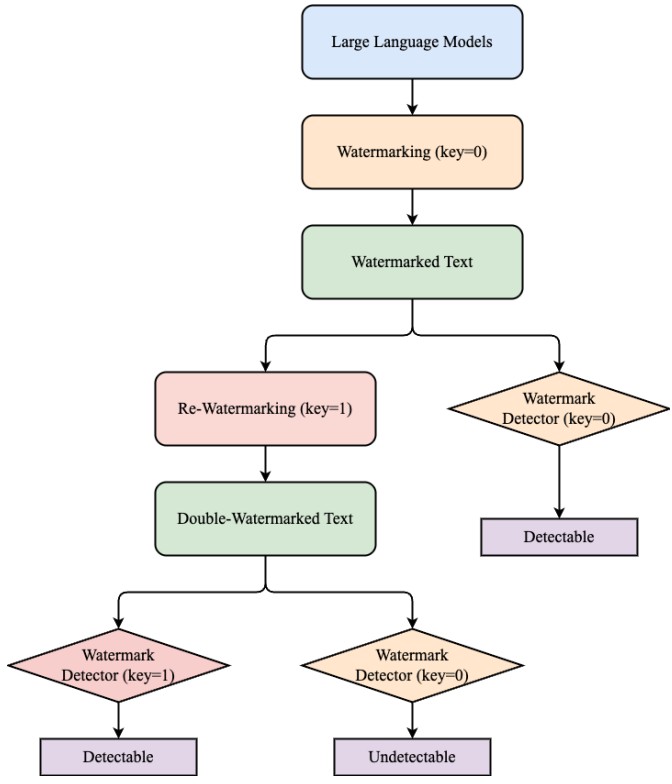

Figure 1: An Overview of the Rewatermarking Attack

**Theorem 1** (Collision-Resistance Implies Overwriting). *Assume both the attacker and the watermark inserter employ the same watermarking method, independently sampling their secret key sequences from some distribution $F \in \Delta(\Xi)^m$. Let $D : \mathcal{W}^* \times \Xi^* \to \{0, 1\}$ be a watermark detection function that satisfies the following collision-resistance-like property:*

$$\Pr_{\{\xi_t\} \sim F} \left[ D(\{w_t\}, \{\xi_t\}) = 1, \right.$$

$$\left. D(\{w_t\}, \{\widehat{\widehat{\xi}}_t\}) = 1 \right] \leq \delta,$$

*for any fixed key sequence $\{\widehat{\widehat{\xi}}_t\}$ and token sequence $\{w_t\}$. Additionally, assume that the watermark mechanism's false negative rate is bounded by $\alpha$.*

*Then, for a text $\{\widehat{w}_t\}$ that is re-watermarked with $\{\widehat{\widehat{\xi}}_t\}$ from an originally watermarked text $\{w_t\}$ (watermarked under $\{\xi_t\}$), the probability that $\{\widehat{w}_t\}$ is* not *detected by the original key $\{\xi_t\}$ is at least $(1-\alpha)(1-\delta)$. Formally,*

$$\Pr \left[ D(\{\widehat{w}_t\}, \{\xi_t\}) = 0 \right] \geq (1 - \alpha)(1 - \delta).$$

Theorem 1 captures a paradoxical phenomenon: a watermark that is *too effective* at distinguishing its own key (i.e., rarely mis-detecting watermarks from other keys) and has a *low false negative rate* (thus easy to embed a watermark) is, in fact, more vulnerable to the re-watermarking attack. The "collision-resistance" property ensures that once a new watermark is successfully added, there is little chance for the original watermark to remain detectable. Meanwhile, a low false negative rate guarantees that adding the new watermark is straightforward. Hence, a highly "accurate" watermark design makes overwriting the old watermark far more likely.

---

**Algorithm 1** Rephrase-based Re-watermarking Attack (with Fine-tuned Rephraser)

---

1: **Input:** Watermarked text $\{w_t\}_{t=1}^n$, fine-tuned rephraser $p_{\text{rephrase}}(\cdot \mid x)$, watermarking mechanism $\mathcal{S}$, new key $\{\widehat{\xi}_t\}_{t=1}^{n'}$
2: **Output:** Re-watermarked text $\{\widehat{w}_t\}_{t=1}^{n'}$
3: **for** $t = 1, 2, \ldots, n'$ **do**
4:     Compute distribution: $P_t = p_{\text{rephrase}}(\{w_s\}_{s=1}^n, \{\widehat{w}_s\}_{s=1}^{t-1})$
5:     Apply watermarking: $\widehat{P}_t = \mathcal{S}(P_t, \widehat{\xi}_t)$
6:     Sample next token: $\widehat{w}_t \sim \widehat{P}_t$
7: **end for**
8: **return** $\{\widehat{w}_t\}_{t=1}^{n'}$

---

## 3 Experiment Results

### 3.1 Experimental Setup

**Tasks and Datasets:** We evaluate the re-watermarking attack across two long-form text generation tasks. The first is *Wikipedia generation*, where we use segments from the Wikipedia corpus to produce continuations of 200+ tokens, following the data processing setup of WikiText-103 Merity et al. (2016). The second is *long-form question answering (QA)*, based on the ELI5 dataset Fan et al. (2019), where models generate detailed answers to complex user queries. These tasks represent realistic and high-stakes scenarios for watermark attribution and content provenance, particularly in the context of misinformation and authorship disputes.

**Paraphraser:** To avoid the semantic similarity loss observed by Luo et al. (2024) during rephrasing, we use a high quality rephraser **StyleRemix** proposed by Fisher et al. (2024) as our rephrasing method. StyleRemix is a state-of-the-art authorship obfuscation tool that perturbs stylistic features while preserving semantic fidelity, making it well-suited for re-watermarking scenarios where altering attribution signals must not degrade meaning.

**Generation Details:** We focus on long text generation scenarios, as prior work Kirchenbauer et al. claims that watermark signals become more robust with increased text length, making this a more challenging and meaningful setting for evaluation. For each prompt, we generate sequences of at least 200 tokens using a watermarked language model. Watermarking is applied using either the Gumbel or Green/Red List scheme with a fixed secret key. To simulate adversarial re-attribution, we rephrase the original generation using StyleRemix while embedding a new watermark under a different key. Some examples of the rewatermarked text are provided in appendix C.

|              | N/A    | No W  | gumbel | gumbel(new) | KGW   | KGW(new) |
|--------------|--------|-------|--------|-------------|-------|----------|
| gumbel(qa)   | 99.84  | 23.53 | 13.51  | 99.55       | 5.13  | 100.00   |
| KGW(qa)      | 100.00 | 67.49 | 50.98  | 100.00      | 37.19 | 100.00   |
| gumbel(wiki) | 99.72  | 63.49 | 36.65  | 99.04       | 0.10  | 99.89    |
| KGW(wiki)    | 100.00 | 90.09 | 77.97  | 99.89       | 61.31 | 99.98    |

Table 1: Results summary for $p \leq 0.01$. Each cell represents the proportion of occurrences satisfying the condition over the total count.

**Detection Results under Controlled False Positive Rate:** Table 1 reports the detection positive rates under various watermarking, rephrasing, and detection settings, with the detection threshold calibrated to control the false positive rate at $p \leq 0.01$. Each row corresponds to the watermarking scheme applied to the original text, where **gumbel** and **KGW** represent the two widely used statistical watermarking methods we consider. "**qa**" refers to the ELI5 dataset, and "**wiki**" denotes the Wikipedia dataset. We also report results with detection thresholds calibrated at $p \leq 0.05$ and $p \leq 0.10$ in Table 5 and 6 in the appendix B.

Each column denotes a different combination of rephrasing strategy and detection key:

- **N/A**: No rephrasing is applied; detection is directly performed using the original watermark detector.

- **No W**: The text is rephrased without applying any watermark; detection is performed using the original watermark detector.

- **gumbel**: The text is rephrased with a new Gumbel watermark; detection is performed using the original watermark detector.

- **gumbel(new)**: The text is rephrased with a new Gumbel watermark; detection is performed using the corresponding new watermark detector.

- **KGW**: The text is rephrased with a new Green/Red list watermark; detection is performed using the original watermark detector.

- **KGW(new)**: The text is rephrased with a new Green/Red list watermark; detection is performed using the corresponding new watermark detector.

**Analysis of Re-watermarking Attack Results.**    We summarize several key observations from Table 1:

First, we find that the newly injected watermark is consistently detectable with extremely high confidence. In all settings, the new watermark achieves detection rates above 99%, indicating that rephrasing with a new watermark reliably embeds a new provenance signal. This contrasts with the findings of Luo et al. (2024), who claimed that in many cases the presence of an existing watermark interferes with the application of a new one. Our results directly challenge this conclusion, demonstrating that **in most cases, new watermarks can be effectively injected**, even when the original text is already watermarked.

Second, we observe that rephrasing with a new watermark consistently leads to a greater degradation of the original watermark's detectability, compared to plain rephrasing without any watermark. For example, using the Gumbel watermark when rephrasing Gumbel-watermarked text reduces the original detector's success rate by approximately 43% on both the QA and Wikipedia datasets compared to plain rephrasing. Moreover, when applying a new Green/Red list watermark to rephrase Gumbel-watermarked text, the detection rate on the QA dataset drops from 99.84% to 5.13%, and on the Wikipedia dataset it drops drastically to just 0.10%. When rephrasing Green/Red watermarked text with a new Green/Red watermark, the detection rate drops by 45% on the QA dataset and by 30% on the Wikipedia dataset.

Interestingly, we also observe that the Green/Red list watermark consistently leads to stronger re-watermarking attacks compared to the Gumbel watermark. This might be because the Green/Red scheme isn't a distortion-free approach, while the Gumbel watermark is (assuming the pseudo-random secret keys used are truly random). Since the Green/Red list scheme allows some bias, it can introduce stronger perturbations during rephrasing, which makes it more likely to erase the old watermark.

### 3.2 Quality of the Rewatermarked Text By Human

In order to reliably study the quality of the rewatermarked text, we not only adopt semantic similarity metrics, but also perform human evaluations by recruiting undergraduate students.

To assess semantic similarity, we first convert the texts into Nomic embeddings by Nussbaum et al. (2024) and then compute the cosine similarity between them. The results, presented in Table 2, show that watermarking has minimal impact on the semantic content of the text. For all tasks and watermarking schemes, the mean cosine similarity exceeds 0.9, indicating a strong preservation of meaning. These findings suggest that the rewatermarked texts generated by StyleRemix remain largely faithful to the original inputs, regardless of the watermarking method applied. Compared with the experiments in Luo et al. (2024), which show that rewatermarking can significantly degrade semantic similarity, sometimes reducing the cosine similarity to around 70% when using methods like green/red lists, we find that a well-designed paraphraser can maintain high semantic fidelity even after rewatermarking. This shows that the reliability of watermarking schemes is even more vulnerable to rewatermark attacks than previously thought, as demonstrated in Luo et al. (2024).

| Task | ReWatermark Type | Mean Similarity | Std. Deviation |
|---|---|---|---|
| Wikipedia (Gumbel WM) | None | 0.9336 | 0.0762 |
| | Gumbel | 0.9149 | 0.0629 |
| Wikipedia (Green/Red WM) | None | 0.9259 | 0.0585 |
| | Green/Red List | 0.9053 | 0.0560 |
| QA (Gumbel WM) | None | 0.9333 | 0.0489 |
| | Gumbel | 0.9328 | 0.0492 |
| QA (Green/Red WM) | None | 0.9259 | 0.0585 |
| | Green/Red List | 0.9194 | 0.0463 |

Table 2: Mean and standard deviation of semantic similarity under different watermarking schemes and tasks (Wikipedia and QA). "WM" denotes the applied watermark (Gumbel or Green/Red List), and similarity is measured using **Nomic embeddings** after **rephrase watermarking (NW)** with **StyleRemix**. Higher mean indicates better semantic preservation.

For the human evaluation, our study involved 33 unique evaluators, primarily undergraduate students, who assessed the similarity between pairs of watermarked and rewatermarked text. Each evaluator was assigned 40 pairs of text for evaluation. However, not all evaluators completed their full set of assignments, resulting in a total of 1014 annotations collected. The evaluations were facilitated through Label Studio, which served as the interface for processing the annotations. The tasks for each evaluator were randomly sampled from a larger pool of prepared tasks to ensure diversity in the evaluations.

Evaluators used a Likert scale to assess similarity between text pairs with the following options: Completely Different, Not Similar, Somewhat Similar, Very Similar, and Exact. To prevent bias, the watermarking and rewatermarking methods were hidden from evaluators. Additionally, evaluators worked in individual projects to avoid influence from other evaluators' annotations. All original text for the pairs was sourced from Wikipedia.

The 40 pairs assigned to each evaluator were evenly distributed as follows:

- 10 pairs: Non-watermarked text rewatermarked by the Gumbel method

- 10 pairs: Gumbel-watermarked text rewatermarked by the Gumbel method

- 10 pairs: Non-watermarked text rewatermarked by the Green/Red List method

- 10 pairs: Green/Red List-watermarked text rewatermarked by the Green/Red List method

Table 3: Human evaluation results with counts shown as absolute values (percentage of row total). Method abbreviations: G = Gumbel, N = None, G/R = Green/Red List.

| Rewatermark | Initial | Different | Not Similar | Somewhat Similar | Very Similar | Exact |
|---|---|---|---|---|---|---|
| G | G | 17 (6.04%) | 24 (8.54%) | 79 (28.11%) | 141 (50.18%) | 20 (7.12%) |
| N | G | 23 (9.31%) | 20 (8.10%) | 61 (24.70%) | 119 (48.18%) | 24 (9.72%) |
| N | G/R | 12 (5.17%) | 27 (11.64%) | 61 (26.30%) | 115 (49.57%) | 17 (7.33%) |
| G/R | G/R | 15 (6.00%) | 36 (14.40%) | 100 (40.00%) | 84 (33.60%) | 15 (6.00%) |

The results of human evaluations are summarized in Table 3. We observe that, regardless of whether Gumbel or Green/Red List rewatermarking is applied, approximately **80%** of the rewatermarked sentences are judged to retain semantic similarity with the original (i.e., labeled as *Somewhat Similar*, *Very Similar*, or *Exact*). **Notably, for Gumbel rewatermarking, 57.3% of the samples are rated as *Very Similar* or above**, whereas **Green/Red List rewatermarking results in 39.6% of samples falling into the same category**. These results indicate that both approaches can achieve moderate to high levels of semantic preservation, though the degree of fidelity preservation differs across watermarking schemes.

In particular, when applying Gumbel watermarking for rewatermarking, we find that although its effectiveness in suppressing the original watermark is relatively lower compared to the Green/Red List watermark,

its impact on semantic preservation is comparable to that of simple rephrasing without watermarking. This suggests that Gumbel watermarking offers a more compatible balance between attribution obfuscation and content fidelity: it achieves excellent semantic preservation, and even if it does not fully eliminate the original watermark, it reliably embeds a new one, thereby introducing ambiguity in content attribution.

## 4 Discussion and Future Work

Our results show that while watermarking is a promising technique for distinguishing AI-generated content from human-written text, its effectiveness in protecting ownership and establishing source attribution remains uncertain. For example, in our rewatermarking experiments, we found that in most cases, a new watermark can be embedded into already watermarked text. As a result, although the original watermark is not always fully erased, the new one frequently remains detectable alongside the old one. This co-existence of multiple watermarks raises a critical question: if both are present, how can we determine which one was embedded first?

An example demonstrating the co-existence of two watermarks is provided below. As shown in Table 4, the algorithm successfully detects both the original and new watermarks simultaneously in this example, confirming that two distinct watermarks can exist concurrently in the same piece of content.

**Original and Paraphrased Text Samples**

**Original Key Generated Text:** *"During a bitter campaign that became known for its mudslinging by both sides, the newspaper's opposition to gambling led it to attack the machine for being controlled by gamblers rather than by businessmen. Nevertheless, it backed the club's nominee for governor, William J. Fields..."*

**Paraphrased Text with New Key:** *"Throughout a contentious electoral contest characterized by widespread mudslinging from both sides, the editorial stance taken by the journal on the topic of gaming resulted in a critique of the machine as ostensibly being governed by gamblers as opposed to businessmen. Nevertheless, the publication continued to support William J. Fields' candidacy for governor..."*

Table 4: P-values of Watermark Detection for the paraphrased text

| Text Version | p-value |
|---|---|
| Original Key | $4.76 \times 10^{-3}$ |
| New Key | $5.71 \times 10^{-20}$ |

In real-world scenarios such as copyright disputes or tracing the origin of AI-generated misinformation, detecting the presence of a watermark alone is insufficient. If two parties can both detect their own watermark on the same piece of content, attribution becomes ambiguous, because current watermarking schemes offer no mechanism to determine which watermark was embedded first. Under this circumstance, it becomes difficult to identify the original source of the content, whether for determining rightful ownership or assigning responsibility for the initial dissemination of misinformation. This remains an open challenge. Future work may include developing methods to determine the embedding order of multiple co-existing watermarks, or designing new watermarking mechanisms capable of explicitly indicating their embedding order.

## 5 Conclusion

In this paper, we revisited the rewatermarking attack proposed by Luo et al. (2024). Through detailed theoretical analysis and comprehensive experimental evaluations, we obtained several key findings that contrast significantly with the conclusions in the original study. Specifically, we demonstrated that: (1) it is feasible to inject a new watermark into already watermarked text in in most cases, (2) these newly injected watermarks effectively overwrite the original watermarks, significantly undermining their detectability, and (3) contrary to previous claims, our rewatermarking method incurs only minimal semantic degradation, as validated

through both embedding-based measurements and extensive human evaluations. These results highlight the previously overlooked threat posed by rewatermarking attacks, calling into question the reliability of existing statistical watermarking techniques.

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

# A  Appendix

## A.1  Proof of Theorem 1

*Proof.* Let $\{\widehat{w}_t\}$ be the text obtained by re-watermarking the originally watermarked text $\{w_t\}$ under the new key sequence $\{\widehat{\xi}_t\}$. By the assumption that the watermark's false negative rate being bounded by $\alpha$, we have

$$\Pr\Big[D\big(\{\widehat{w}_t\},\{\widehat{\xi}_t\}\big) \,=\, 1\Big] \,\geq\, 1 - \alpha.$$

In other words, since $\{\widehat{w}_t\}$ is indeed watermarked by the new key $\{\widehat{\xi}_t\}$, the probability of detecting that watermark using the new key is at least $1 - \alpha$.

Next, by the collision-resistance-like property, whenever $D(\{\widehat{w}_t\},\{\widehat{\xi}_t\}) \,=\, 1$, the probability that $D(\{\widehat{w}_t\},\{\xi_t\}) = 1$ also holds is at most $\delta$. Equivalently,

$$\Pr\Big[D\big(\{\widehat{w}_t\},\{\xi_t\}\big) = 1 \,\Big|\, D\big(\{\widehat{w}_t\},\{\widehat{\xi}_t\}\big) = 1\Big] \,\leq\, \delta,$$

which implies

$$\Pr\Big[D\big(\{\widehat{w}_t\},\{\xi_t\}\big) = 0 \,\Big|\, D\big(\{\widehat{w}_t\},\{\widehat{\xi}_t\}\big) = 1\Big] \,\geq\, 1 - \delta.$$

We can now chain these two facts together. We have

$$\Pr\Big[D\big(\{\widehat{w}_t\},\{\xi_t\}\big)=0\Big]$$

$$\geq \ \Pr\Big[D\big(\{\widehat{w}_t\},\{\xi_t\}\big)=0 \ \wedge \ D\big(\{\widehat{w}_t\},\{\widehat{\xi}_t\}\big)=1\Big]$$

$$= \ \Pr\Big[D\big(\{\widehat{w}_t\},\{\xi_t\}\big)=0 \ \Big| \ D\big(\{\widehat{w}_t\},\{\widehat{\xi}_t\}\big)=1\Big]$$

$$\times \Pr\Big[D\big(\{\widehat{w}_t\},\{\widehat{\xi}_t\}\big)=1\Big]$$

$$\geq \ (1-\delta)(1-\alpha).$$

Thus, with probability at least $(1-\delta)(1-\alpha)$, the re-watermarked text $\{\widehat{w}_t\}$ *fails* to be detected under the original key $\{\xi_t\}$, meaning the old watermark is effectively overwritten. $\qquad\square$

## B   Watermark Detection

We report the results under the same experiment setting as in Table 1. Specifically, Table 5 presents results obtained with a detection threshold of 0.05, while Table 6 reports results for a detection threshold of 0.1.

|  | N/A | No W | gumbel | gumbel(new) | KGW | KGW(new) |
|---|---|---|---|---|---|---|
| gumbel(qa) | 99.85 | 35.96 | 22.79 | 99.61 | 12.10 | 100.00 |
| KGW(qa) | 100.00 | 80.86 | 67.75 | 100.00 | 60.45 | 100.00 |
| gumbel(wiki) | 99.86 | 75.57 | 50.42 | 99.54 | 20.02 | 99.93 |
| KGW(wiki) | 100.00 | 95.41 | 87.81 | 100.00 | 78.73 | 99.98 |

Table 5: Results summary for $p \leq 0.05$. Each cell represents the proportion of occurrences satisfying the condition over the total count.

|  | N/A | No W | gumbel | gumbel(new) | KGW | KGW(new) |
|---|---|---|---|---|---|---|
| gumbel(qa) | 99.85 | 44.22 | 29.94 | 99.61 | 17.76 | 100.00 |
| KGW(qa) | 100.00 | 87.45 | 75.75 | 100.00 | 70.54 | 100.00 |
| gumbel(wiki) | 99.90 | 81.33 | 58.39 | 99.68 | 28.69 | 99.96 |
| KGW(wiki) | 100.00 | 97.04 | 91.19 | 99.94 | 85.65 | 99.98 |

Table 6: Results summary for $p \leq 0.1$. Each cell represents the proportion of occurrences satisfying the condition over the total count.

## C   Paraphrase Examples

In Table 7, we present several examples of pairs consisting of the original watermarked texts and their paraphrased versions. We observe that among these examples, although the paraphrased texts closely preserve the original meanings, the original watermark becomes undetectable after rephrasing.

| Original Watermarked Text | Paraphrased Text (New Watermark) | p-value (Original Key) |
|---|---|---|
| Gorman and the producers of the show were concerned that viewers might be turned away from what they considered an excessively violent scene. However, upon seeing a test edit of a scene where an eye is removed from a victim with a knife, Abrams felt it needed more "violence and horror", which he felt the audience wanted. To ensure accuracy in their portrayal of science and technology, producers worked with the Science Advisory Committee of Fringewho were comprised of scientists from various organizations, including the MIT Haynes Institute for Data Science (now the Haystack Observatory). As a result of their work with this group, Fringingained the reputation of having the most scientifically sound depiction of technology of all science-fiction television shows... | In the context of the show's producers and their concerns regarding potentially excessive violence depicted therein, the producers sought a test edition of a sequence wherein a victim's eye was extracted by a knife, which raised concerns among the producers and their advisory team for the need of increased violent and horrific scenes as desired by the audience. To guarantee the precision of scientific and technological representation, the producers collaborated closely with a Science Advisory Committee from Fringes comprising eminent scientific experts from various organizations, including the MIT Haynes Institute for Data Science — subsequently renamed the Haystack Observatory. These collaborations resulted in the acquisition of an enviable reputation for Fringe as having the most authentic depiction of technology across all science-fiction television shows... | 0.36473 |

| | | 0.23664 |
|---|---|---|
| From March until May, Japanese warships attacked and captured a number of islands in Melanesia. Allied resistance on the islands was minimal and Allied leaders believed these islands were too isolated for Allied defence. Source: Department of Veteran Affairs. Retrieved September 2007. In the months preceding the fall of Singapore in February 1922 Australian soldiers and airmen were deployed in numbers that were unprecedented for that nation's military. More than 5,500 Australian servicemen, including a complete division of ground soldiers, were dispatched from their home country for deployment with the Malayan and British Indian Armies on what is now the Malay peninsula in Southeast Asia. Despite their military service in a region with a much different culture and environment from that with which they were most familiar, many Australian personnel found the experience of their service there both fascinating and rewarding. One Australian officer in particular was able to share this fascination with readers of The Sydney Mail, Australia' s most popular weekly magazine at that time. On the evening of Saturday 13 April 1934, Major-General Cyril Brudenell White VC presented the first annual award for military journalism at a function in Sydney. His decision to award a prize for journalism recognised the role the civilian media has traditionally played in shaping the relationship between the military and civilian populations of our nation. For over a hundred years, our military personnel, both active and reserves, have regularly taken time from military training and preparation for overseas deployments to assist the communities that they are based around. | In the period preceding the fall of Singapore in 1922, an unprecedented number of Australian soldiers and airmen were mobilized for deployment as part of the Malayan and British Indian Armies on the Malay peninsula in Southeast Asia. These deployments consisted of more than 5,500 Australian servicemen, including an entire infantry division, representing a considerable commitment from the country. Although these deployments required serving overseas in a significantly different cultural and environmental context from their native surroundings, the experiences were found by many Australian personnel to be intriguing and profoundly satisfying. A notable instance of this was the publication by a particular Australian officer in the popular Australian publication 'The Sydney Mail', a renowned magazine of that era. On the 13th of April 1934, the Major-General Cyril Brudenell White VC delivered the inaugural award presentation for military journalism at a ceremony hosted in Sydney. His decision to institute an award in recognition of journalism reflected a longstanding recognition of the significant impact that civilian media outlets have historically exerted on the dynamic between our military and civilian populations within the country. Over the past century and a half, our active and reserve military personnel have regularly taken leave from their training and preparation for overseas deployments in order to assist the local communities they serve. These engagements reflect... | |

| | | |
|---|---|---|
| There are several small lakes in or at the foot of various can-yons; most are ephemeral. A large portion is part of a Bureau wildlife area, but there is little or no recreational use because the terrain in most areas consists of dry shrublands with poor footing for hikers and horses. Location and description Edit Located about southeast of Burns in north-central Oregon in Malheur County, the southern part is the most prominent part with the north part much more subdued and only a bit higher than the nearby high deserts of Malhaur and Harney counties. In the westernmost area there are high ridgeline in a narrow section of hills that are the only part over 6000 feet elevation with a large lake at an elevation of over four thousand feet and a small lake about three thousand five hundred feet. The highest area has a prominent high point and the area to its north is a hump with high points at both ends and then a more rolling area with numerous high hills. At about a thousand elevation lower there's a major canyon with many tributaries and numerous small and large ponds. To the northeast of there there's a very high highpoint and several highpoints of at least a couple thousand higher. South of it there also is high area that is relatively narrow with higher hills at each extreme but is very humped with elevation differences in it of several thousand. Much of Nevada's portion consists mostly rolling terrain. Vegetation and wildlife Edit... | There exist a series of small lakes nestled within various canyon bottoms; the vast majority exhibit ephemeral characteristics. An ample segment constitutes a component of a Bureau-designated wildlife region, however, the region sees minimal to no recreational usage due to the predominance of arid chaparral landscapes rendering the grounds unsuitable for hiking or horse riding. Location and Descriptive Features The region in question lies approximately southeast of Burns in the north-central region of Oregon within Malheur County. Notable for its elevated prominence is the southern segment, whereas the more subdued northern segment stands at a mere fraction of the elevation observed throughout the Malhaur and Harney counties deserts. Within the far west region, an elevated ridge-line extends in a narrow strip of hills representing the region's highest elevations exceeding 6000 feet; a sizable lake at approximately 4000 feet as well as another smaller body of water at approximately 3500 feet constitute the region. Moving east, the region features an elevated highpoint surrounded by a humped region containing several more elevated hills before transitioning to a more gently undulating terrain characterized by many elevated hills. Substantially reduced by 1000 feet lower in elevation is an expansive and multi-fingered canyon system accompanied by several small and larger ponds. Moving east further still and significantly elevated from the region, several towering elevated highpoints ranging from multiple thousands of feet higher can be found. Similarly located to the south of this region, another... | 0.72575 |

| | | |
|---|---|---|
| A machine gunner from Company A who saw the attack reported it and alerted the nearby gunners of Battery D from 1/11. Within a minute of their report the guns were firing at point blank range at a range of only 100 yards (91 m). As a result of this quick action and with their line of approach exposed by the flash and sound, about two hundred of Kokusko's soldiers fell before reaching Edsons ridge. In the morning light, it became evident that the attacking unit was a detachment of a Japanese regiment under Major Mochitsura Hashimoto that were not part of Kawguichis brigade and were unaware of its plan of attack, leading them astray from their original objective. Although the unit's loss was severe—almost a quarter of Hashimotos detachment were killed—Kawaguchis attack plan was not seriously affected because the detachment's mission was only to screen the main attack by Kokubunis unit from a counterattack by a U.S. force from Taivoa Point. Kokubuni continued to attack with a second wave of soldiers, but the first wave's heavy losses from Marine machine guns led to its failure to achieve any breakthrough into Edsos positions. Edson organized a rear defense line and moved the survivors of Companies I and K back into an area on Hill 123, where they were supported with mortars and machine guns. | An alert was promptly issued by a Company A machine gunner witnessing the attack, which promptly notified the nearby gunners from Battery D of the 1/11th Infantry Battalion. In a mere minute following this alert, the guns began firing at a dangerously close range of approximately 100 yards (approximately 91 meters). Due to this swift response and the exposure of the attackers' line of advance by the sound and light, around two hundred of Kokuskos troops perished before approaching Edsos ridge.

Subsequent observations in the light of day revealed the identity and composition of the attacking unit: a detachment of a Japanese regiment led by Major Mochitsura Hashimoto who were not affiliated with Kawaguchis' brigade and were therefore unfamiliar with its attack strategy; consequently, this led them astray from their original objective.

Although the detachment suffered significant casualties, amounting to almost a quarter of the detachment strength perished in the attack, the impact on Kawaguchis' overall attack plan remained minimal, as their objective was merely to distract Kokubuni's unit from any potential counterattack from U.S. forces stationed at Taivoa Point.

Kokubuni persisted with a subsequent wave of troops in the attack, however, the significant casualties among the first-wave troops inflicted by the Marines' formidable machine gun fire rendered the attempt at a breakthrough futile.

Edson devised a rear defense line while transferring... | 0.10654 |

Table 7: Comparison between the original text with KGW watermark and paraphrased text containing a new KGW watermark. The p-value was calculated using the KGW detecting algorithm with the key originally used to generate the watermark in the original text.

