# OpenReview forum: "Erasing and Tampering Statistical Watermarks via Re-watermarking in Large Language Models"
_TMLR — Withdrawn by Authors_

### Review · Reviewer_RCYS · 2025-10-23

**Summary Of Contributions:**

In this work, the authors revisited the attack on the watermarked text by watermarking it with Gumbel-watermark or Green/Red List watermark. For the experiments, the authors evaluate the semantic similarity of the generated text after the attack first by the cosine similarity and then by undergraduate students annotations. In the end, the authors concluded that it may be feasible inject a new watermark into already watermarked text, which deteriorates the detectability of the original watermark.

**Audience:**

No

**Audience Explanation:**

Please look at the comment for the section "Are the claims made in the submission supported by accurate, convincing and clear evidence?".

To sum up, although the technical part looks correct, the results are trivial. Besides, the experimental results do not look convincing to me.

**Claims And Evidence:**

Yes

**Claims Explanation:**

Overall, the paper is written clearly, and the technical/math part of the paper is correct, which consists only of one Theorem 1 on the lower-bound on probability detection of watermark in rewatermarked text. However, the result of the theorem is trivial, and the conclusion that a higher false negative rate for original watermark detection leads to more vulnerabilities to the attacks via re-watermarking is intuitive.

Regarding the semantic similarity:
- Cosine similarity between texts is a bad metric for semantic similarity.
- Were the undergraduate students given examples of texts that are "Different", ..., "Very Similar", "Exact" (this is not mentioned in paper at all)? It is an important part to make sure that the grading scale of each person is at least kind of similar. Otherwise, different people can have a view on the scale of the assessment.
- How many students assess each pair of texts? It is usually required that each sample be reviewed by multiple evaluators.
- I respectfully disagree that the rewatermarking method incurs only minimal semantic degradation. As it was reported, only 57% and 40% of the text pairs fell into "Very Similar" or higher category, which looks like on very high percentage.

**Requested Changes:**

Please see the comments above.

---

### Review · Reviewer_GCg1 · 2025-11-16

**Summary Of Contributions:**

This paper proposes a new re-watermarking attack that is highly effective in removing existing statistical watermark from LLM models. The main contributions of this paper are:
1. Identify that with a high performing rephraser, new watermark can be reliably inserted into an LLM with already existing watermark
2. Show both emperically and theoreitically that newly injected watermark can overwrite the exisiting watermark in the model
The observation of the paper provides an effective, low-cost method for removing watermark from protected LLMs.

On the weakness side, the method lacks ablation on the rephrasing technique used and on the effect against different base LLM models. More details in the requested change part.

**Audience:**

Yes

**Audience Explanation:**

LLM copyright is an emerging research topic with rising interests. This work is interesting to both practitioners and researchers.

**Broader Impact Concerns:**

As an attack paper, it would be interesting to know if the attack method has a significant impact on the current adoption practice of LLMs in personal and business uses. A broader impact statement is needed.

**Claims And Evidence:**

Yes

**Claims Explanation:**

Adequate experimental results are provided to show that new watermarks are effectively inserted into the model, and the insertion indeed weakens the detectability of the original watermark significantly. Experiments are done under diverse combinations of watermarking techniques.

On the other hand, the evidence can be further improved by showing the impact of different rephrasing techniques to the final attakc success rate.

**Requested Changes:**

1. The key of the proposed method is the use of the rephraser. The rephrasing model, StyleRemix, is treated as a blockbox tool and assumed to be always effective. More discussion is needed on the impact of rephrasing quality and rephrasing model choice to the final success rate of the proposed method. Alternative rephrasing attacks like structural text changes, translation-reversion attacks, or more adaptive rephrasing methods could further strengthen the generalizability of the findings against different watermark families.

2. The paper does not make clear discussion on the base LLM model used for the watermark experiment. It would be interesting to see if the proposed method can be effective across multiple base pretrained LLM models.

---

### Review · Reviewer_Mev8 · 2025-11-17

**Summary Of Contributions:**

This paper revisited the robustness issue of re-watermarking attack based on text rephrasing. Contradicting to previous work by Luo et al, this work found that with the help of a high quality rephrasing model, re-watermarking can retain the semantic similarity. Furthermore, the new watermark can either overwrite or co-exist with the existing ones, which greatly compromises the efficacy of the original watermark. This finding highlights that the risk of re-watermarking attack could be much higher than previously known and calls for further studies on enhancing the LLM watermarking mechanism.

**Audience:**

Yes

**Audience Explanation:**

LLM data source and copyright issue has been a topic of great interests.

**Claims And Evidence:**

Yes

**Claims Explanation:**

The effect of rewatermarking is experimentally shown in Table 1 and 3. The semantic similarity is maintained as shown in Table 2.

**Requested Changes:**

**1. The choice of rephrasing models**

The author claimed that the main reason this work differed from Luo's work is the use of a high quality rephrasing model. It would be more complete and convincing to support this claim with another set of data using a different (lower quality) rephrasing model in Section 3. In addition, the author may want to elaborate a little more on the differences between this rephrasing model, StyleRemix, and other commonly used general purpose LLMs, e.g. Llama. For example, readers may wonder if the selection of the model is based on output quality or a trade-off with inference efficiency?

**2. Theorem 1 vs experimental data**

Theorem 1 suggests that a very specific detector (when the probability "lower case delta" is very low) would be more vulnerable to re-watermarking attack, as the chance this detector "fails to find" the original watermark in the new re-watermarked text is very high, i.e. the old watermark is considered fully erased. However, Table 1 shows that it's quite often that this "lower case delta" probability is far from being small, which undermines the value of Theorem 1. In fact, as the author pointed out later in Section 4, as long as the new watermark can be successfully added and detected, it doesn't matter whether the old watermark is fully erased or not. The goal of rewatermarking attack can still be considered achieved. It would be better to mention this alternative success criteria much earlier in the manuscript so that the readers can keep both in mind while digesting the experimental data.

**3. Minor issues, such as typos.**

- Page 3 Section 1.1, Paragraph 3 says "A watermarking mechanism S is ..." but next paragraph says "Given a language model p and a watermarking mechanism W...". The second occurrence seems to be a typo.

- Table 1, all the Gumbel are lower case. Would be better to be consistent with other uses in the manuscript.

- p-value in Table 4 seems to have a different definition compared to that in Table 1. Author may want to clearly state the definition and meaning in Section 4 paragraph 2 or the caption of Table 4 to avoid confusions.

---

### Note · Authors · 2025-11-29

I have read and agree with the venue's withdrawal policy on behalf of myself and my co-authors.